# Beyond `[cls]`: Exploring the true potential of Masked Image Modeling representations

**Marcin Przewięźlikowski**[1,2,*]    **Randall Balestriero**[3]    **Wojciech Jasiński**[1,4]
**Marek Śmieja**[1]    **Bartosz Zieliński**[1]

[1]Faculty of Mathematics and Computer Science, Jagiellonian University
[2]Doctoral School of Exact and Natural Sciences, Jagiellonian University
[3]Brown University    [4]AGH University of Krakow
[*]`marcin.przewiezlikowski@doctoral.uj.edu.pl`

## Abstract

Masked Image Modeling (MIM) has emerged as a promising approach for Self-Supervised Learning (SSL) of visual representations. However, the out-of-the-box performance of MIMs is typically inferior to competing approaches. Most users cannot afford fine-tuning due to the need for large amounts of data, high GPU consumption, and specialized user knowledge. Therefore, the practical use of MIM representations is limited. In this paper we ask what is the reason for the poor out-of-the-box performance of MIMs. Is it due to weaker features produced by MIM models, or is it due to suboptimal usage? Through detailed analysis, we show that attention in MIMs is spread almost uniformly over many patches, leading to ineffective aggregation by the `[cls]` token. Based on this insight, we propose Selective Aggregation to better capture the rich semantic information retained in patch tokens, which significantly improves the out-of-the-box performance of MIM [1] [2].

## 1   Introduction

Self-supervised Learning (SSL) [9] has emerged as a powerful paradigm for pre-training visual representations from unlabeled data. These representations are of high quality and can be used out-of-the-box for various downstream tasks [29, 16, 38, 4]. There are two dominant SSL paradigms: Joint Embedding Architectures (JEA), which optimize the goal of producing similar embeddings from multiple views of the same image [30, 18, 20, 15, 58, 28, 21, 22, 16, 60, 38], and Masked Image Modeling (MIM), which learns to reconstruct missing pixels (or high-level representations) of images with occluded fragments [29, 6, 56, 11, 43, 4]. Although MIM is an arguably more generic pretext task, requiring fewer assumptions about the pretraining data, the resulting representations often underperform JEAs in high-level perceptual tasks for reasons not fully understood [59, 40, 10].

In this paper, we systematically analyze how masked models form their representations in order to understand the reasons for their poor quality. We find that MIM representations do not work well with the two standard ViT feature extraction methods – the `[cls]` tokens and average patch representations, which are commonly treated as global image descriptors [25, 16, 29]. This is because, unlike JEAs, MIM representations are ineffective at aggregating the relevant semantic information (see left and center in Fig. 1), which contributes to the performance gap between these two approaches.

These findings lead us to propose **Selective Aggregation** of MIM patch representations as a remedy. Using a lightweight technique inspired by Multiple-Instance Learning [37], we consistently improve

---

[1]The full version of this paper has been previously published at ICCV 2025.
[2]We release the codebase at `github.com/gmum/beyond_cls`.

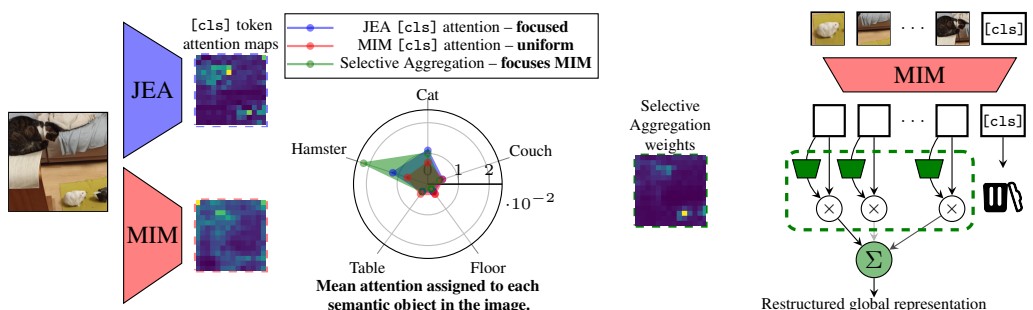

Figure 1: ViTs trained with Joint-Embedding Architectures (JEA) attend to semantically rich patches while forming global [cls] representations, which is critical for perception performance. At the same time, ViTs trained with Masked Image Modeling (MIM) attend more uniformly to all patches, absorbing both relevant and irrelevant information and achieving an effect similar to naive average pooling (see **left** and **center**). To improve out-of-the-box MIM performance, we propose Selective Aggregation (see **right**) – a mechanism that aggregates patch tokens according to their relevance, as quantified by a lightweight linear regressor (■).

the quality of representation for a wide range of MIM models without fine-tuning their parameters The improvements resulting from Selective Aggregation in the well-established [29, 56] and recently published [4, 24] models support the key finding that the lack of proper aggregation is an inherent problem in MIMs. With the continued emergence of novel approaches [24], we expect Selective Aggregation to remain a useful tool for their developers and users.

## 2 Analysis of information flow in MIM and JEA

The [cls] token in Masked Image Models (MIMs) captures a representation that can, to some degree, serve as a global image descriptor [29, 56]. However, its out-of-the-box quality is significantly lower than the [cls] token obtained from Joint-Embedding Architectures (JEAs), limiting the effectiveness of standard probing techniques. This raises the question: *What are the differences in how the [cls] tokens gather information in these two approaches?* Understanding these differences will allow us to build a deeper understanding of JEA and MIM vision transformers and, in consequence, develop a principled approach to feature extraction. To this end, we study their self-attention mechanism, as it is the only means by which the [cls] token acquires information from the image patches. For a preliminary on vision transformers and MIM, we refer to Appendix B.

**Methodology.** In self-attention, each token either recycles its representation by attending to itself or gathers the representations of other tokens by attending to them. In Fig. 2, we analyze these interactions to understand how information flows between [cls] and patch tokens in publicly available ViTs pretrained with several popular SSL approaches [16, 22, 60], including the most popular MIM – the Masked Autoencoder (MAE) [29]. Specifically, we measure: (in Fig. 2a) the proportion of attention the [cls] token assigns to itself, and (in Fig. 2b) the entropy of [cls] attention to the patch tokens.

**Key findings.** Our analysis reveals significant differences in how information is exchanged between tokens of JEA- and MAE-trained ViTs. The [cls] token in JEA strongly attends to patch tokens, allowing it to integrate relevant information across ViT blocks. In contrast, the MAE [cls] token heavily recycles its representation, limiting its ability to aggregate new information. Moreover, the remaining attention of the MAE's [cls] token is almost uniformly distributed across all patch tokens, potentially absorbing redundant or irrelevant information. JEA models exhibit lower entropy, meaning their [cls] tokens focus on fewer, more important patches. Crucially, fine-tuning MAE for classification shifts the attention of [cls] and patches closer to that of JEA, highlighting the importance of selective attention in forming strong representations.

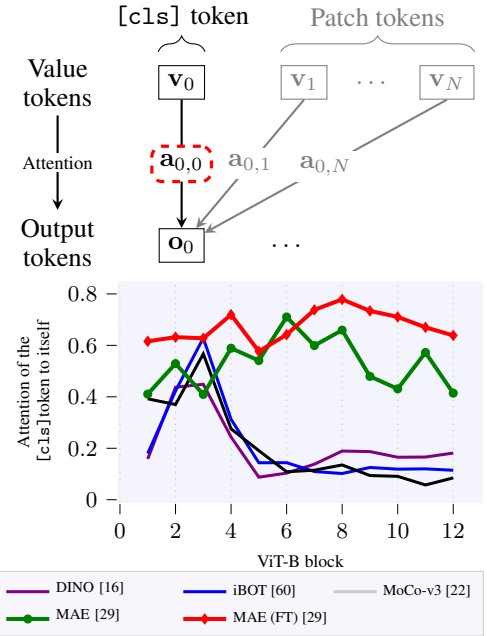 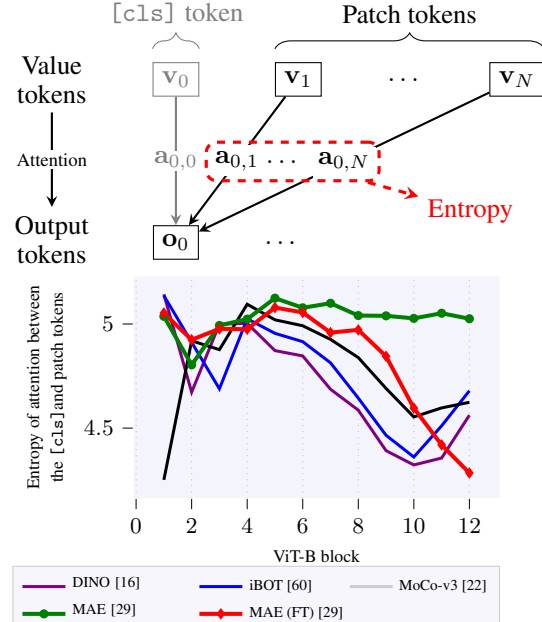

(a) Attention of the [cls] token to itself is much higher in MAE, than in the JEA ViTs. As opposed to JEA, where the [cls] tokens gather a large amount of information from the patch tokens, the MAE [cls] tokens primarily recycles its own representation.

(b) Entropy of [cls] token attention to patch tokens reaches almost the maximal possible level in MAE[3]. In other models, it decreases in the deeper model blocks, indicating that the [cls] token attends to different patches in a more selective manner. Fine-tuning of MAE decreases this entropy, indicating that selective attention to patch tokens is crucial for good perception.

Figure 2: Analysis of the self-attention mechanism of the [cls] tokens in ViTs trained via different SSL techniques. In the graphs in the upper part of the figure, $\mathbf{v}_i$ denote the value projections, $\mathbf{a}_{0,i}$ are the attention weights between the [cls] and patch tokens, and $\mathbf{o}_i$ – self-attention outputs.

## 3 Selective Aggregation of Masked Image Modeling representations

Our analysis showed that masked models do not form structured global representations as effectively as JEA models because their [cls] tokens do not properly aggregate high-level information from the relevant patches. This leads us to ask: *Can we improve the quality of the MIM representation simply by modifying its aggregation scheme?*

To address this, we propose **Selective Aggregation**, a mechanism that dynamically assigns importance to tokens when forming the final representation. Specifically, we define an aggregation function $s : \mathbb{R}^{N \times D} \to [0,1]^N$ that predicts a score vector $\mathbf{s} \in [0,1]^{N+1}$ weighting patch tokens from the $L$-th ViT encoder block $\mathbf{z}_{L,1:N} \in \mathbb{R}^{N \times D}$ in a summation-based aggregation mechanism [8]. The weights of $\mathbf{s}$ identify the key tokens and aggregate them into the representation $\mathbf{z}_{\text{select}} = \sum_{i=0}^{N} \mathbf{s}_i \mathbf{z}_{L,i} \in \mathbb{R}^D$, which can then be used as a drop-in replacement for the [cls] token or the naively averaged representation. The existence of a function $s$ that aggregates tokens into a representation better than the [cls] token would indicate that the MIM patch tokens actually contain high-level information that has not been captured by [cls], supporting our hypothesis that MIM models do not naturally form structured global representations.

We implement Selective Aggregation with Attention-based Multiple Instance Learning Pooling (AbMILP) [37] – an approach that dynamically assigns importance weights to tokens, enabling structured aggregation while maintaining minimal complexity. Given a set of vectors (in our case, tokens $\mathbf{z}_L$), AbMILP predicts aggregation weights by applying a linear model $t : \mathbb{R}^D \to \mathbb{R}$ to each

---

[3]The theoretical upper bound of [cls]-patch entropy is 5.27 for a discrete distribution over 196 patches.

Table 1: Linear probing accuracy on ImageNet-1k [46] for different global image representations. In Masked Image Models, patch tokens aggregated via Selective Aggregation consistently produce global representations of higher quality than those obtained from the `[cls]` and naively averaged patch tokens.

| | Encoder | | Representation aggregation method | | | |
|---|---|---|---|---|---|---|
| | Source | ViT | Avg. pooling of patches | `[cls]` token | *Selective (ours)* patches | + `[cls]` |
| **Masked Image Modeling** | MAE [29] | ViT-S | 47.1 | 47.4 | **54.4** | **54.6** |
| | MAE [29] | ViT-B | 65.8 | 67.8 | **71.6** | 71.5 |
| | MAE [29] | ViT-L | 73.0 | 75.8 | **77.4** | 77.4 |
| | MAE [29] | ViT-H | 73.8 | 77.0 | **78.1** | 78.0 |
| | SimMIM [56] | ViT-B | 54.3 | 51.5 | **62.8** | 62.0 |
| | MaskFeat [53] | ViT-B | 56.9 | 62.9 | **66.6** | 65.8 |
| | BEIT-v2 [43] | ViT-B | 78.5 | 78.9 | **80.9** | 81.0 |
| | I-JEPA [4] | ViT-H | 77.7 | – | **79.2** | - |
| | CAPI [24] | ViT-L | 76.2 | – | **82.4** | - |
| **JEA** | iBOT [60] | ViT-B | 75.0 | 77.8 | 77.9 | **78.2** |
| | DINO-v2 [38] | ViT-B | 81.9 | 83.2 | **83.5** | 83.5 |
| | DINO [16] | ViT-B | 71.1 | **76.6** | 75.2 | 76.2 |
| | MoCo-v3 [22] | ViT-B | 71.1 | **75.1** | 75.1 | 75.2 |
| | MAE (+ FT) [29] | ViT-B | 76.6 | **80.0** | 79.1 | **79.8** |

vector, followed by softmax:

$$\mathbf{s}_i^{\text{AbMILP}} = \frac{\exp(t(\mathbf{z}_{L,i}))}{\sum\limits_{j=0}^{N} \exp(t(\mathbf{z}_{L,j}))}.$$

Crucially, Selective Aggregation only restructures the existing out-of-the-box ViT representations without transforming them into a different representation space. This ensures that our evaluation isolates the impact of aggregation itself, without modifying confounding factors such as the inherent quality of MIM token representations [10, 2]. From a practical standpoint, this allows for a lightweight implementation of the aggregation function, requiring only a single linear regressor to assign token weights. As a result, the additional computational and parameter overhead is negligible compared to the base ViT.

## 3.1 Evaluation of Selective Aggregation in high-level perception tasks

We evaluate how Selective Aggregation affects the global representations of vision transformers. We follow the MAE linear probing protocol [29], described in detail in Appendix C.

We first evaluate the quality of representations formed by the `[cls]` token, average patch representation, and Selective Aggregation on the task of ImageNet-1k classification [46]. To understand the effect of Selective Aggregation, we apply it to a wide selection of prominent MIM and JEA models in two variants: **(i)** aggregating only the patch tokens, and **(ii)** aggregating the patch and the `[cls]` tokens[4]. We report the results in Tab. 1. We observe consistent improvements in a wide variety of MIMs which were pretrained with both low-level [29, 56, 53]), and high-level [43, 4, 24] prediction targets. This supports our hypothesis that the lack of such aggregation is an inherent problem in MIMs, regardless of how they are trained. On the contrary, in JEAs, Selective Aggregation and the `[cls]` representations have similar quality, confirming that these models can select the relevant patches out-of-the-box. Aggregating the `[cls]` token with patches is insignificant in MIMs, further confirming its low representation quality.

Having established that Selective Aggregation improves MIM performance, in Tab. 2 we further evaluate it with several MIM models on the more challenging low-shot, fine-grained, and multilabel classification tasks. The favorable performance of Selective Aggregation further reinforces its usefulness.

---

[4]I-JEPA [4] and CAPI [24] do not include the `[cls]` tokens in their architecture.

Table 2: Evaluation of standard ([cls] for all models, except for I-JEPA [4] and CAPI [24]), and selectively aggregated MIM representations on low-shot (ImageNet-1% [46]), fine-grained (CUB [52], Stanford Cars [34], OxfordIIIPets [41], Food-101 [14]), and multilabel (NUS-WIDE [23], COCO [36]) classification tasks. Selective Aggregation consistently improves or matches MIM performance on these tasks.

| Encoder | | ImageNet-1% [46] | | CUB [52] | | Stanford Cars [34] | | OxfordIIIPets [41] | | Food-101 [14] | | NUS-WIDE [23] | | COCO [36] | |
| Source | ViT | Standard | Selective | Standard | Selective | Standard | Selective | Standard | Selective | Standard | Selective | Standard | Selective | Standard | Selective |
|---|---|---|---|---|---|---|---|---|---|---|---|---|---|---|---|
| MAE [29] | ViT-B | 39.1 | **48.3** | 45.8 | **65.9** | 31.8 | **58.8** | 82.4 | **90.8** | 68.1 | **78.0** | 67.2 | **67.9** | 61.2 | **65.3** |
| SimMIM [56] | ViT-B | 17.3 | **34.5** | 17.9 | **61.8** | 11.3 | **23.5** | 37.9 | **47.7** | 54.2 | **61.9** | 58.8 | **60.0** | 41.5 | **44.9** |
| BEIT-v2 [43] | ViT-B | 66.8 | **69.0** | 79.2 | **80.4** | 72.2 | **74.9** | **93.7** | 93.4 | 88.2 | **90.2** | 69.5 | **71.9** | 71.4 | **76.9** |
| I-JEPA [4] | ViT-H | 66.4 | **70.9** | 51.7 | **59.9** | 40.9 | **42.7** | 89.9 | **92.4** | 81.1 | **85.1** | 71.7 | **72.2** | 70.7 | **73.6** |
| CAPI [24] | ViT-L | 52.7 | **74.2** | 25.9 | **79.7** | 45.6 | **76.8** | 83.8 | **94.5** | 85.9 | **90.5** | 71.8 | **73.3** | 72.3 | **80.1** |

# 4 Conclusion

Masked Image Models (MIMs) are increasingly popular, yet their out-of-the-box usefulness in high-level perception tasks is suboptimal. This paper presents an in-depth analysis of why that is the case. We analyze the attention of [cls] token for various SSL approaches and conclude that MIMs attend more uniformly to all patches when producing global representation. In contrast, better-performing Joint-Embedding Architectures (JEAs) are more selective and, as a result, accumulate only relevant information. As a remedy, we propose Selective Aggregation of the patch representations returned by MIM. We demonstrate that this approach consistently improves the perception performance of multiple MIM models, regardless of whether their original prediction target was low-level pixels or high-level latent representations. These results support the hypothesis that a proper aggregation of the information stored in the patch tokens is crucial for high-quality representations in vision transformers. We hope that this new perspective on MIM representations will inspire future work on improving these models, and pave the way for their broader practical applications.

## Acknowledgements

This research has been supported by the flagship project entitled "Artificial Intelligence Computing Center Core Facility" from the Priority Research Area DigiWorld under the Strategic Programme Excellence Initiative at the Jagiellonian University. The research of Marcin Przewięźlikowski was supported by the National Science Centre (Poland), grant no. 2023/49/N/ST6/03268. The research of Wojciech Jasiński was supported by the National Science Centre (Poland), grant no. 2022/47/B/ST6/03397. The research of Marek Śmieja was supported by the National Science Centre (Poland), grant no. 2022/45/B/ST6/01117. The research of Bartosz Zieliński was supported by the National Science Centre (Poland), grant no. 2023/50/E/ST6/00469. We gratefully acknowledge Polish high-performance computing infrastructure PLGrid (HPC Center: ACK Cyfronet AGH) for providing computer facilities and support within computational grant no. PLG/2024/017148. We thank Marcin Sendera, Adam Pardyl, Turhan Can Kargin, Bill Psomas, Michał Pietruszka, and Klaudia Bałazy for fruitful discussions and feedback over the course of this work.

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

# Appendix

## A  Related works

**Self-supervised learning (SSL) of visual representations**   has become a cornerstone of modern computer vision, enabling models to learn without labeled data [1, 9]. Several powerful SSL paradigms have been developed, including Joint-Embedding Architectures (JEA) [30, 18, 15, 16, 38], which learn representations by enforcing invariance across augmented image views, leading to strong out-of-the-box performance on high-level tasks. However, JEA approaches rely on carefully designed data augmentations [47] and implicitly assume similar distributions between pretraining and downstream data [5, 38], limiting their adaptability [54, 35, 17, 44, 49, 26]. As an alternative, Masked Image Modeling (MIM) [50, 51, 6, 29, 56, 24] reconstructs masked image regions or their representations, leveraging Transformers' ability to model long-range dependencies [29, 43, 39]. This paradigm has demonstrated strong fine-tuning performance and scalability [29, 57, 4, 45], motivating further study into how MIM models structure information and how their representations can be effectively utilized [59, 40, 10]. Our work investigates this problem by analyzing how MIM models structure information and identifying a crucial shortcoming in their attention mechanisms.

**Differences in representation structure between JEA and MIM**   have been the subject of several studies analyzing their attention patterns and feature organization [59, 33, 40, 10]. JEA models are known to produce compact, global representations, often relying on the `[cls]` token to aggregate features [16, 58]. In contrast, prior work has shown that MIM models tend to focus on local structure [40, 32, 55], leaving open the question of how their learned representations interact across tokens and how suitable they are for typical probing strategies in downstream tasks. Rather than directly addressing these differences, recent works propose to probe ViTs with additional attention layers [19, 12] containing significantly more trainable parameters. However, the reason why such complex probing is needed remains unexplored. Our work fills this gap by systematically analyzing the information flow in ViTs pretrained with JEA and MIM, uncovering previously overlooked fundamental structural differences between both paradigms. Furthermore, we show that these differences contribute to inefficiencies when using MIMs for high-level perception tasks, highlighting the need for a lightweight probing approach that accounts for the lack of appropriate representation structure in MIMs.

## B  Preliminaries

In this section, we recall the basic Vision Transformer (ViT) architecture [25], and the Masked Autoencoder (MAE) [29] – the most popular Masked Image Modeling technique.

### B.1  Vision transformers (ViT)

**Image processing by ViT**   begins by dividing and flattening an image $\mathbf{x} \in \mathbb{R}^{H \times W \times C}$ into a sequence of $N$ non-overlapping *patches* $\mathbf{x}_p \in \mathbb{R}^{N \times (P^2 \cdot C)}$, where $(P, P)$ is the resolution of a patch and $N = \frac{HW}{P^2}$. Next, a linear projection layer $e : \mathbb{R}^{(P^2 \cdot C)} \to \mathbb{R}^D$ transforms each patch into a $D$-dimensional embedding to which appropriate positional encoding vectors $\mathbf{p} \in \mathbb{R}^{N \times D}$ [25] are added. We refer to the result of these operations as *patch tokens*:

$$\mathbf{z}_p = e(\mathbf{x}_p) + \mathbf{p} \in \mathbb{R}^{N \times D}. \tag{1}$$

We also define a learnable [cls] token $\mathbf{x}_{cls} \in \mathbb{R}^D$, which is prepended to $\mathbf{z}_p$[5]. The first ViT block input is defined as:

$$\mathbf{z}_0 = [\mathbf{x}_{cls}; \mathbf{z}_p] \in \mathbb{R}^{(N+1) \times D} \tag{2}$$

The $l$-th ViT block transforms tokens $\mathbf{z}_{l-1}$ into tokens $\mathbf{z}_l$. Each of the $L$ blocks is a sequence of Multihead Self-Attention (MSA) [48] and MLP layers. For both MSA and MLP, the input is first normalized with LayerNorm [7], and the output of the layer is summed with the unnormalized input, forming a residual connection [31].

**Multihead Self-attention (MSA)** [48] is a key component of ViT, which allows for exchanging image information between tokens. It consists of $h$ self-attention heads, each of which separately transforms the sequence of $(N+1)$ input tokens into a sequence of output tokens of the same length. A self-attention head creates three linear projections of the input, $\{\mathbf{q}, \mathbf{k}, \mathbf{v}\} \in \mathbb{R}^{(N+1) \times (D/h)}$ and computes the self-attention map $\mathbf{a} \in [0,1]^{(N+1) \times (N+1)}$:

$$\mathbf{a} = softmax(\frac{\mathbf{q}\mathbf{k}^T}{\sqrt{D/h}}), \tag{3}$$

Output tokens $\mathbf{o} \in \mathbb{R}^{(N+1) \times (D/h)}$ are calculated as $\mathbf{o} = \mathbf{a}\mathbf{v}$, i.e. the sums of $\mathbf{v}$ weighted by subsequent rows of $\mathbf{a}$. Next, the output tokens of each self-attention head are concatenated along their token dimension and projected through a linear layer to form the final output of the MSA.

**Final vision transformer representation** $\mathbf{z}_L$ consists of $(N+1)$ tokens of shape $D$. In high-level perception tasks such as image classification, the most common strategy is to use only the [cls] token output of the final ViT block ($\mathbf{z}_{L,0}$) as the representation of the entire image which serves as an input to the classifier [25, 16, 59]. The same approach is used in JEA pretraining, where the invariance objective is imposed on the [cls] representations (typically followed by a projector network [18, 13]), while patch tokens are discarded [16, 22]. An alternative strategy is to summarize the image representation as the average value of patch tokens, i.e. $\sum_{i=1}^{N} \frac{\mathbf{z}_{L,i}}{N}$, sometimes even removing the [cls] token from the model [29, 3]. However, this typically leads to representations of worse quality [25].

## B.2  Masked Image Modeling

Masked Image Modeling (MIM) [50, 51] is a paradigm of learning representations through the task of image inpainting (masking random contents of images and training a model to reconstruct them). This approach is straightforward to apply in vision transformers because masking can be implemented by randomly removing a subset of patch tokens. Among the various MIM implementations [56, 6], the Masked Autoencoder (MAE) [29] has emerged as one of the most popular frameworks.

**Masked Autoencoder (MAE)** consists of two ViTs – an encoder $f$ and decoder $g$. During MAE pretraining, we divide the image into patch tokens $\mathbf{z}_p$, remove a random subset of tokens, and then process the remaining ones through the encoder. The tokens to be removed are selected by a random binary mask $m \in \{0,1\}^N$, where 0 is drawn with the probability of $\rho$ (mask ratio) and denotes the dropped tokens. In consequence, the input and output sequences of $f$ consist of $(1 + N \cdot (1 - \rho))$ tokens (the [cls] token and $N \cdot (1 - \rho)$ patch tokens).

Before processing the output of $f$ through the decoder[6] $g$, we complement it with $N \cdot \rho$ identical *mask tokens* $\mathbf{z}_{msk} \in D$, such that the placement of mask tokens reflects the placement of tokens removed by mask $m$. The decoder adds an appropriate positional embedding to both, encoded and mask tokens. After obtaining the output sequence of $g$, we discard the [cls] token and project the $N$ patch tokens into the sequence $\hat{\mathbf{x}}_p \in \mathbb{R}^{N \times (P^2 \cdot C)}$, i.e. of the same size as the image patches $\mathbf{x}_p$.

---

[5]For convenience of notation, the [cls] token will have the index of 0, and patch tokens will have the indices $\in 1...N$.

[6]For simplicity of notation, we assume that the encoder and decoder have equal embedding sizes and numbers of layers, denoted by $D$ and $L$, respectively. In practice, if the embedding sizes are not equal, we prepend the decoder with an appropriate linear projection.

The objective function of MAE is defined as the mean squared error between the image pixels and predicted pixels, calculated at the patches that were randomly dropped by mask $m$:

$$\mathcal{L}_{MAE} = \mathbb{E}_\mathbf{x} ||\mathbf{x}_p[1-m] - \hat{\mathbf{x}}_p[1-m]||^2. \tag{4}$$

Numerous works propose to replace the MAE prediction target with higher-level representations of patches. Such targets can be formed from low-variance image components [53, 10], or latent representations of an image encoder [11, 43, 57, 4, 24]. However, the reconstruction objective is typically applied to the mask tokens, whereas the `[cls]` representation does not optimize any objective. This raises the question of what representation is formed by `[cls]` token, and whether it is the optimal choice for a global descriptor in high-level perception tasks.

## C Detailed experimental setup

In this section, we describe our experimental methodology: our choice of pretrained models, the details and hyperparameters of evaluating their representations, as well as the codebase used for the experiments.

### C.1 Overview of the analyzed vision transformers

Our study aims to verify whether Selective Aggregation of patch token representations with AbMILP can yield form better representations than those of the `[cls]` tokens.

For this purpose, we analyze various vision transformer architectures that were pretrained with several MIM and JEA approaches, using the parameters shared by the authors of the respective methods. This has two advantages:

- Using the existing parameters significantly reduces the computational resources required for our study.

- Our study provides insights about the *very same* sets of parameters that are described in their respective literature and used by the wider research community.

For a fair evaluation, we use the parameters of the models that were pretrained on the ImageNet-1k dataset [46]. All of the explored model parameters are compatible with the implementations of the MAE [29] or SimMIM [56] vision transformers. Following the MAE and DINO implementations, when using ViT-S and ViT-B, we split the image into a $14 \times 14$ grid of patches of size $16 \times 16$. When using ViT-H, the we split the image into $16 \times 16$ patches of size $14 \times 14$.

The only analyzed models that are not publicly available but were trained by us are the ViT-S pretrained with the MAE and the fine-tuned ViT-S/B/L variants of the MAE. To prepare these models, we used the MAE pretraining and fine-tuning codebase and hyperparameters [29]. Before fine-tuning, we initialize the model with the pretrained MAE parameters as shared by the authors and use the `[cls]` token representation as input to the classifier.

### C.2 Representation evaluation details

In our evaluation of ViT representations in terms of classification accuracy on ImageNet-1k, we follow the MAE linear probing protocol [29]: we augment the images only by random cropping, use the batch size of 16,384, and train the classifier head for 90 epochs (50 in the case of ViT-Large and Huge) with the LARS optimizer [27], the base learning rate of 0.1 with cosine decay and 10 epochs of warmup, optimizer momentum of 0.9, and no weight decay. For CUB and ImageNet-1%, we follow a similar linear probing setup but train using SGD with a batch size of 1024. We report the results averaged over 3 random seeds. When using the AbMILP Selective Aggregation, we train it alongside the classifier head.

These evaluations are performed on a single node equipped with 4 NVIDIA-GH200 GPUs. Due to the memory constraints of this setup, we obtain the effective batch size of 16,384 by aggregating gradients from two forward passes with half of that batch size.

Table 3: Comparison of AbMILP [37] and Attentive Probing (AP) [19] aggregation schemes. AbMILP and the single-head cross-attention AP perform comparably.

| Encoder | | Aggregation method | | |
|---|---|---|---|---|
| Initialization | ViT type | AbMILP | AP (1 head) | AP (12 heads) |
| MAE [29] | ViT-S | 54.4 | 53.6 | **63.9** |
| MAE [29] | ViT-B | 71.6 | 71.4 | **75.4** |
| MAE [29] | ViT-L | 77.4 | 77.6 | **79.7** |
| MAE [29] | ViT-H | 78.1 | 78.3 | **80.0** |
| BEIT-v2 [43] | ViT-B | 80.9 | 81.0 | **81.8** |
| I-JEPA [4] | ViT-H | 79.2 | 79.5 | **79.7** |
| CAPI [24] | ViT-L | 82.4 | 81.6 | **82.7** |

## C.3 Codebase

Our code is based on the official MAE codebase [29], written in PyTorch [42], and available at github.com/gmum/beyond_cls. We include scripts required for the analysis of the attention mechanism in ViTs, as well as linear evaluation of their representations extended with AbMILP [37].

## D   Selective Aggregation and Attentive Probing

Attentive Probing (AP) [19] has been proposed as an alternative to naive feature aggregation in ViTs. Similarly to our Selective Aggregation, AP learns to emphasize the most relevant patch tokens while keeping the encoder parameters frozen. However, AP differs from our approach in a key way: it does not only learn to aggregate tokens, but also transforms them with a cross-attention layer into a new representation space. potentially more suitable for the downstream task [10]. In contrast, AbMILP is designed to isolate the aggregation process while preserving the original ViT representations.

We compare AP and AbMILP across multiple MIM models in terms of ImageNet-1k classification and report the results in Tab. 3. Since AP typically uses a 12-head self-attention mechanism, we additionally evaluate a reduced variant with a single attention head (without reducing the representation dimensionality) to better compare with the capacity of AbMILP (which predicts a single set of representation aggregation weights). As expected, the full AP model achieves the best results, benefiting from its greater expressive power. However, despite AP's significantly higher parameter and compute cost, reducing it to a single head brings its performance in line with AbMILP. This result is somewhat surprising and suggests that AP's strength may come from ensembling multiple Selective Aggregation patterns rather than from the learned transformation. Exploring this insight to develop more efficient Selective Aggregation strategies is a promising direction for future work.

