# OpenReview forum: "Beyond [cls]: Exploring the true potential of Masked Image Modeling representations"
_NeurIPS.cc/2025/Workshop/UniReps — UniReps2025_

### Official Review · Reviewer_224P · 2025-09-10

**Confidence:** 4

**Review:**

This work investigates the performance gap between JEA and MIM models. The authors find that JEA models learn more focused CLS token representations by selectively aggregating informative patch token representations, whereas in MIM methods, the attention masks of the CLS token representations uniformly aggregate over patch tokens. Inspired by AbMIL, the authors propose Selective Aggregation as a remedy and demonstrate that this approach provides an efficient way of closing the gap between JEA and MIM.

# Strengths
- The authors tackle a highly relevant problem within the SSL literature and propose a surprisingly simple way to improve MIM methods without fine-tuning the full pretrained model.
-  Experiments are convincing and cover a broad range of current SOTA methods.
- The manuscript is well written, easy to follow, and contains illustrative figures.
# Weaknesses
- The biggest weakness of this paper is the fairness of the evaluation protocol. The authors train a linear probe on top of $z_{select}$, where $z_{select}$ is either average pooling over all tokens, $z_{CLS}$, or by applying the Selective Aggregation. However, while in the first two cases, the authors actually train a linear probe, Selective Aggregation effectively learns a non-linear model due to the softmax in the formulation, arguably leading to comparability issues. It might be more fair to compare a linear probe with Selective Aggregation to the other two aggregation methods with a one hidden layer MLP.
- Besides the AbMIL-based aggregation, it might be interesting to see whether attention pooling could lead to even better results.

**Score:**

4

**Topic Fit:**

2

---

### Official Review · Reviewer_ABJh · 2025-09-13
**The objective is interesting, existing experiments are smart but should be more, writing should be improved**

**Confidence:** 4

**Review:**

Summary:

The paper is addressing why masked image modeling methods perform inferior than JEPA competitors when evaluated in the frozen setup.

Strenghts:

1 - The idea of measuring the entropy of attention maps with the CLS token is interesting. The experiment that MIM models attend uniformly to all the patches instead of focusing on target areas is insightful. As well as, the conclusion that uniform attention will result in the uniform aggregation of both relevant and irrelevant information features, resulting in low performance in frozen evaluation.

2 - The experiment in Table 1 where it is shown that the performance for aggregated patches < cls + aggregated patches for MIM models, supports the goal of the paper, showing a good aggregation is helpful.

3 - The experiments on fine-grained datasets look strong, emphasizing the method's extra significance on fine-grained tasks

Weaknesses:

1 - The paper proposed a new way of aggregation based on assigning a score to each patch and taking it as the weight of aggregation. This is done through a linear score prediction. I can not find in the paper if this linear layer is trained or not. If it is trained, then it is not fair to compare it with frozen evaluation peformance and if it not trained, then how is it initilized? is it random? if yes, then I am wondering why it can perform better than the CLS. If we assume the uniform attention issue is a justified assumption, then CLS should also perform as some sort of random weight, which makes it similar to the proposed approach. The paper need to clarify this section sufficiently.

2 - I suggest adding attentive probing results to the paper as well. In attentive probing, a few attention blocks are learned instead of linear layers, which has similar goal to what is discussed in the paper. The method should distinguish its pros and cons against attentive probing through more experiments.

**Score:**

2

**Topic Fit:**

2

---

### Official Review · Reviewer_1hPu · 2025-09-15
**Distinguishing between training objective, and selective aggregation of patch token, and other forms of evaluation tasks**

**Confidence:** 3

**Review:**

The authors introduce a method that learns a linear weighting of image patch tokens. When combined with MIM models, this approach improves performance on image classification.
While the method highlights the difference between MIM and JEA models as well as the importance of feature aggregation in downstream tasks, I believe the authors should also consider the role of the training process in different models, along with carefully designing other forms of evaluation when reporting their results. Below, I tried to expand on these:

Models with a masking paradigm are trained to estimate the distribution of the training dataset in the pixel space (or token space). Essentially, they learn many auto-regressive models (https://arxiv.org/pdf/1502.03509v2) that can predict the value of missing pixels (tokens), using a combination of others, and nothing more. In their pretrained version, they have no bias towards the accurate estimation of global shape features and structures. In contrast, JEA models learn similar latent embeddings for different views of the same image, implicitly, learning features (through self-distillation) that are shared across the two views and are higher-level than pixel values. Considering these arrangements of training, I think it is expected to see a uniform attention assigned to objects in MIM models, and a non-uniform one for JEA models (Figure 1). This is supported by the result in Table 1: the linear probing accuracy of a fine-tuned MAE is significantly higher than that of pretrained MAEs with Selective Aggregation, and is comparable to JEA models. This means that, through fine-tuning, the attention is turned toward the features in the image that are useful for classification.

 This brings me to the second point of my review, which is the evaluation task. As far as I understand, the models were evaluated on image classification (i.e., object recognition). What JEA models learn is aligned with the object recognition objective. They implicitly represent information about shape and global structure during training, while MIM models do not. Thus, specifically evaluating for an object recognition task may be somewhat unfair to MIM models. It would be interesting to see how the method performs on other downstream tasks beyond classification.
That said, I think the paper introduces an important characteristic about different architectures, and I would look forward to seeing a complete version of the extended abstract with additional analyses.

**Score:**

3

**Topic Fit:**

2

---

### Official Review · Reviewer_Uawr · 2025-09-16
**Not enough experimental validation**

**Confidence:** 5

**Review:**

In this work, the authors compare the cls token of ViT models trained against a learnable sum of the patch embeddings. From experimental validation, they try to prove that their learnable sum works better than the patch embeddings.

I like how simple the work is.

Limitations:

1. Needs more experimental validation on hard zero/FSL datasets, performance under Distribution shift, OOD detection, pretraining, and other tasks. Needs at least 3-4 more such tables showing the efficacy and robustness of their approach against the traditional [cls] and aggregation methods. What if I want to take a current pretrained ViT model and want to fine-tune it on a new dataset with this approach? Will it give any beneficial results? Also, I understand that Table 2 address fine-grain and FSL classification, but I don't think that's enough.

2. Using patches embeddings directly, even if they have a learnable aggregation function, might make the model really vulnerable to spurious correlations. How does this work do in those situations? Can the authors perhaps add some experiments for that? Perhaps the WaterBirds dataset? A good problem to solve will be how to learn this [cls] aggregation funciton while staying robust to spurious correlations and also beating the traditional [cls] token?

**Score:**

1

**Topic Fit:**

3